# A regressive analysis of the main environmental risk factors of human echinococcosis in 370 counties in China

Liying Wang[1,2,3], Zhiyi Wang[1], Min Qin[1], Jiaxi Lei[1], Xixi Cheng[1], Jun Yan[4]*, Laurent Gavotte[3], Roger Frutos[2]

1 National Institute of Parasitic Diseases, Chinese Centre for Disease Control and Prevention; Chinese Centre for Tropical Diseases Research; WHO Collaborating Centre for Tropical Diseases; National Centre for International Research on Tropical Diseases, Ministry of Science and Technology; Key Laboratory of Parasite and Vector Biology, Ministry of Health, Shanghai, People's Republic of China, 2 Cirad, UMR 17, Intertryp, Montpellier, France, 3 Espace-Dev, University of Montpellier, Montpellier, France, 4 Chinese Centre for Disease Control and Prevention, Beijing, China

* yanjun@chinacdc.cn

**Data Availability Statement:** The data used for this publication are considered pseudonymized personal data. As such, the "The Data Security Law

## Abstract

### Background

Echinococcosis is a natural focal, highly prevalent disease in China. Factors influencing the spread of echinococcosis are not only related to personal exposure but also closely related to the environment itself. The purpose of this study was to explore the influence of environmental factors on the prevalence of human echinococcosis and to provide a reference for prevention and control of echinococcosis in the future.

### Methods

Data were collected from 370 endemic counties in China in 2018. By downloading Modis, DEM and other remote-sensing images in 2018. Data on environmental factors, *i.e.*, elevation, land surface temperature (LST) and normalized difference vegetation index (NDVI) were collected. Rank correlation analysis was conducted between each environmental factor and the prevalence of echinococcosis at the county level. Negative binomial regression was used to analyze the impact of environmental factors on the prevalence of human echinococcosis at the county level.

### Results

According to rank correlation analysis, the prevalence of human echinococcosis in each county was positively correlated with elevation, negatively correlated with LST, and negatively correlated with NDVI in May, June and July. Negative binomial regression showed that the prevalence of human echinococcosis was negatively correlated with annual LST and summer NDVI, and positively correlated with average elevation and dog infection rate. The prevalence of human cystic echinococcosis was inversely correlated with the annual average LST, and positively correlated with both the average elevation and the prevalence rate of domestic animals. The prevalence of human alveolar echinococcosis was positively

of the People's Republic of China" and "The Management Measures for Data Sharing and Use of the China Center for Disease Control and Prevention (2022 Edition)" apply to the use of this dataset. For any request for access to data, the applicant can contact: National Institute of Parasitic Diseases, Chinese Centre for Disease Control (Chinese Centre for Tropical Diseases Research), available at phone number: 021-64377008. The applicant is informed that access and reuse of this dataset will require obtaining (i) submitting an application through the website of the Public Health Science Data Center (https://www.phsciencedata.cn/Share/), (ii) obtaining approval from the National Institute of Parasitic Diseases, Chinese Centre for Disease Control, (iii) signing the "China Center for Disease Control and Prevention Data Use Responsibility Agreement" and "Data Sharing Use Agreement", which stipulate service content, intellectual property ownership, confidentiality requirements, usage responsibility, breach of contract responsibility, dispute resolution, and other matters.

**Funding:** The funding support for this study comes from The National Natural Science Foundation of China (Grant No. 81703281) and NHC Key Laboratory of Echinococcosis Prevention and Control, China (No.2021WZK1006) The funders had no role in study design, data collection and analysis, decision to publish, or preparation of the manuscript.

**Competing interests:** The authors have declared that no competing interests exist.

correlated with both NDVI in autumn and average elevation, and negatively correlated with NDVI in winter.

## Conclusion

The prevalence of echinococcosis in the population is affected by environmental factors. Environmental risk assessment and prediction can be conducted in order to rationally allocate health resources and improve both prevention and control efficiency of echinococcosis.

## Author summary

Echinococcosis is a natural focal, highly prevalent disease in China. Factors influencing the spread of echinococcosis are not only related to personal exposure but also closely related to the environment itself. This study covered all endemic counties in China and studied the main environmental factors affecting the epidemic of echinococcosis. to explore the influence of environmental factors on the prevalence of human echinococcosis, and to provide a reference for prevention and control of echinococcosis in the future. Data were collected from 370 endemic counties in China in 2018. By downloading Modis, DEM and other remote-sensing images in 2018. The prevalence of human echinococcosis in each county was positively correlated with elevation, negatively correlated with LST, and negatively correlated with NDVI in May, June and July. Prevalence of human echinococcosis was negatively correlated with annual LST and summer NDVI, and positively correlated with average elevation and dog infection rate. Prevalence of human cystic echinococcosis was inversely correlated with the annual average LST, and positively correlated with both the average elevation and the prevalence rate of domestic animals. The prevalence of human alveolar echinococcosis was positively correlated with both NDVI in autumn and average elevation, and negatively correlated with NDVI in winter.

## Introduction

Echinococcosis, also known as hydatidosis, is a zoonotic parasitic disease caused by the larvae of *Echinococcus* which is of worldwide concern. Two types of human echinococcosis that are currently prevalent in China, namely, cystic echinococcosis (CE) which is caused by the larvae of *Echinococcus granulosus* and alveolar echinococcosis (AE) which is caused by the larvae of *Echinococcus multilocularis* [1]. China is one of the countries with the highest prevalence in the world, mainly in pastoral and semi-pastoral areas in the west and north, leading to detrimental impacts on the health of local populations and socio-economic development [2]. 40% of CE cases and 91% of AE cases worldwide occur in China where the disease burden of CE and AE accounts for 40% and 95% of the global disability-adjusted life years (DALYs), respectively [2–3]. In 2018, 47,278 echinococcosis cases have been recorded in China with 44,730,268 people being exposed over 370 endemic counties from 10 provinces or autonomous regions. All 370 counties were endemic for CE while 115 were also endemic for AE [4].

The prevalence of echinococcosis in China displays a strong spatial autocorrelation with a spatial distribution depending upon geographical, meteorological, biological and socio-economic factors [4–5]. Because it is a natural focal disease, its prevalence is closely related to the local natural environment. Environmental and ecological factors play a crucial role in the life cycle of *Echinococcus*, and these environmental factors can affect the spread of echinococcosis

to humans [5]. Landscape factors are an important driving force for the spread of echinococcosis [6]. For example, elevation is positively correlated with the prevalence of CE [7–8]. The transmission cycle of *E. multilocularis* involves wild animals, and small rodents as intermediate hosts. The distribution of small mammals is related to the natural land cover and changes will affect their habitat [9]. The risk of human AE is related to the population density of small mammals [10–11]. Environmental factors are directly or indirectly influencing the survival rate of *Echinococcus* eggs, the distribution of wild animal population, the spatial distribution of echinococcosis and the risk of disease in the human population [12]. Most researches on influencing factors focus on human behavior. Environmental factors commonly used in echinococcosis studies include elevation, temperature and vegetation index.

This study covered all endemic counties in China and studied the main environmental factors affecting the epidemic of echinococcosis, including elevation, land surface temperature (LST) and normalized difference vegetation index (NDVI). By using univariate rank correlation analysis and fitting county-level negative binomial regression model, we aimed to understand the effects of elevation, LST, and NDVI on the incidence of echinococcosis, providing reference for better targeted prevention and control measures and rationally allocating health resources.

## Materials and methods

### Ethics statement

This survey was approved by the Ethical Review Committee of the National Institute of Parasitic Diseases, Chinese Center for Disease Control and Prevention (No. 20160810). The performed activities were all within the scope of the national project for echinococcosis control. All participants were informed of the content and purpose of the investigation and examinations, potential complications, consequences as well as benefits before examination. Those who agreed to participate were required to sign written informed consent forms. All participants were given feedback. All echinococcosis diagnosed patients provided written agreements to participate and were provided with free drug treatment or subsidized surgical costs.

With the progress of the Central Government's Transfer Payment Project for Echinococcosis Control (CGTPPEC), each county carried out population screening with a coverage rate of over 90%. The incidence rate of echinococcosis was assessed in each township (town, street) by abdominal ultrasound examination of local residents of 2 years old and above. The diagnosis was done according to WS 257 standards [13] based B-ultrasound assisted serological examination. The infection rate of *Echinococcus* in dogs was assessed by necropsy or arecoline hydrobromide catharsis on more than 20 dogs, including domestic and stray dogs, in each administrative village/community. In villages/communities with less than 20 dogs, all dogs were tested. Detection in dogs is mainly carried out through fecal antigen detection reagents [14]. Prevalence data and dog infection rates from the 370 epidemic counties were obtained from the annual report system of the Annual Task of CGTPPEC in 2018.

### Acquisition of environmental data

Elevation, LST and Normalized Difference Vegetation Index (NDVI) were extracted as follows. For average elevation data in each county, SRTM3 data with an accuracy of 90m covering the whole area were downloaded. MODIS data MOD11A2 were downloaded from the NASA website (https://search.earthdata.nasa.gov), range and time were set and data were filtered for downloading. Satellite image cloud removal was performed and global data were synthesized. Mosaic, projection, splicing, cutting, data inspection and zonal statistics were then carried out to extract monthly LST in each epidemic county. NDVI is a quantitative index of vegetation

coverage and characteristics of vegetation changes. NDVI data require MODIS Reprojection Tool (MRT) for formatting and projection conversion. In order to eliminate the influence of outliers, the maximum composite method (MVC) was used to synthesize NDVI data, and the maximum monthly NDVI image was used to characterize the vegetation coverage. Furthermore, seasonal and annual average LST and NDVI were calculated as follows: spring LST and NDVI refer to the average in March, April and May; summer LST and NDVI refer to the average in June, July and August; autumn LST and NDVI refer to the average in September, October and November; and winter LST and NDVI refer to the average in December, January and February. A comprehensive information database was then integrated and constructed.

## Statistical analysis

The normality of the prevalence distribution of echinococcosis at the county level was determined by the Shapiro-Wilk test and its correlation with relevant environmental variables was analyzed. If the distribution was normal, the Pearson correlation coefficient was then used to describe the correlation. If the data did not follow a normal distribution, the Spearman rank correlation coefficient was used. Data in this study displayed a variance higher than the mean, a non-randomness and a spatial autocorrelation of the distribution. Therefore, they were fitted with a negative binomial distribution. The negative binomial model is a generalized linear model with logarithmic links yielding binomial random variables [15–16]. In order to test the over dispersion hypothesis and the preference of negative binomial model compared with Poisson model, a Lagrange multiplier test was used. The Proc Genmod program in SAS9.1 (SAS Institute, Cary, NC) was used to model the prevalence at the county level (natural logarithm) by negative binomial regression. Normality test and rank correlation analyses were performed on IBM SPSS 19.0 (Statistical Package for the Social Science). A $P$ value lower- than 0.05 was defined as statistically significant.

## Results

### Normality of the distribution

The Shapiro-Wilk test showed that the prevalence rate of echinococcosis at the county level did not follow a normal distribution. The W value was 0.428 ($p<0.05$), which was a normal skew distribution, as shown in Fig 1.

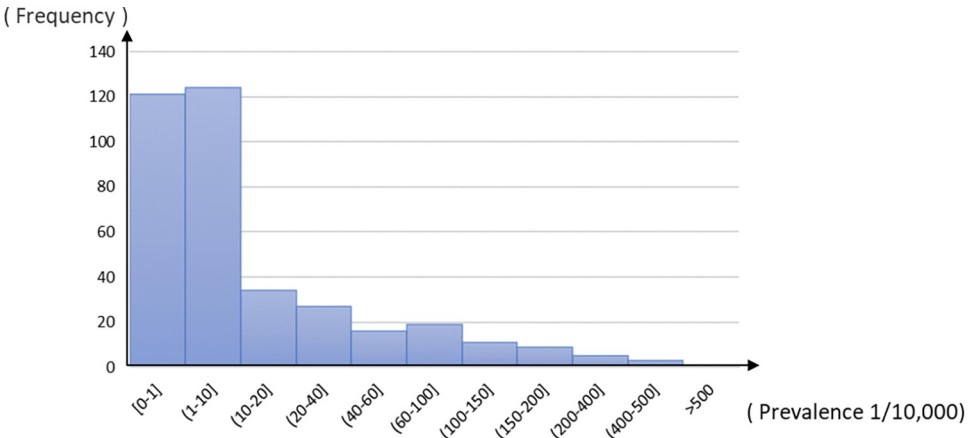

**Fig 1. Frequency distribution histogram of echinococcosis prevalence.**

**Table 1. Seasonal distribution of LST in endemic counties.**

| LST (˚C)/Season | Spring (Number of counties) | Summer (Number of counties) | Autumn (Number of counties) | Winter (Number of counties) |
|---|---|---|---|---|
| < -10 | 0 | 0 | 0 | 14 |
| -10 ≤ LST < -5 | 0 | 0 | 0 | 48 |
| -5 ≤ LST < 0 | 0 | 0 | 0 | 104 |
| 0 ≤ LST < 5 | 14 | 0 | 24 | 158 |
| 5 ≤ LST < 10 | 108 | 8 | 133 | 16 |
| 10 ≤ LST < 15 | 97 | 68 | 100 | 16 |
| 15 ≤ LST < 20 | 98 | 90 | 100 | 0 |
| 20 ≤ LST < 25 | 39 | 53 | 9 | 0 |
| 25 ≤ LST < 30 | 0 | 76 | 0 | 0 |
| LST ≥ 30 | 0 | 61 | 0 | 0 |
| Total | 356 | 356 | 356 | 356 |

The highest LST was 39.47˚C in Shanshan County, Turpan, Xinjiang in July, and the lowest LST was -27.68˚C in Chenbarhu Banner, Hulun Buir, Inner Mongolia in January. The LST range was described in groups. The LST distribution in the 370 endemic counties is shown in Table 1 and Fig 2. The lowest NDVI was 0.00015 in Fuhai County, Altay Prefecture, Xinjiang in January while the highest NDVI was 0.86 in Daguan County, Zhaotong City, Yunnan in July. The NDVI range was described in groups. The distribution of NDVI in 370 endemic counties in each province is shown in Table 2 and Fig 3.

## Rank correlation between environmental variables and prevalence of echinococcosis at the county level

Rank correlation analysis was carried out between LST and NDVI of each month, quarter and annual average and the prevalence rate of county-level population, as shown in Table 3. When

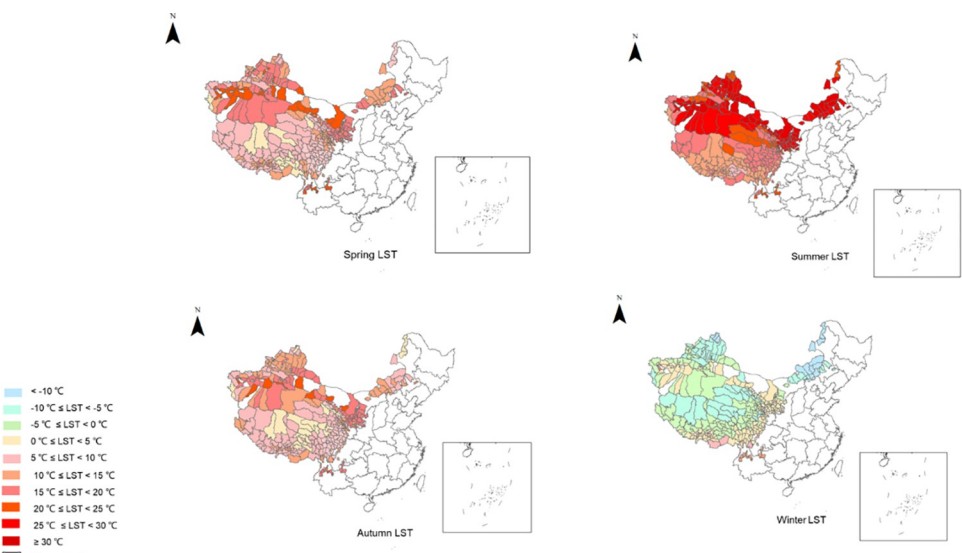

**Fig 2. Seasonal LST distribution in endemic counties.** The base layer is from https://www.webmap.cn/mapDataAction.do?method=forw&resType=5&storeId=2&storeName=%E5%9B%BD%E5%AE%B6%E5%9F%BA%E7%A1%80%E5%9C%B0%E7%90%86%E4%BF%A1%E6%81%AF%E4%B8%AD%E5%BF%83&fileId=BA420C422A254198BAA5ABAB9CAAFBC1 with credit to National Catalogue Service For Geographic Information.

**Table 2. Seasonal NDVI distribution in endemic counties.**

| NDVI /Season | Spring (Number of counties) | Summer (Number of counties) | Autumn (Number of counties) | Winter (Number of counties) |
|---|---|---|---|---|
| < 0.1 | 79 | 22 | 39 | 135 |
| $0.1 \leq$ NDVI $< 0.3$ | 194 | 95 | 164 | 160 |
| $0.3 \leq$ NDVI $< 0.5$ | 62 | 96 | 88 | 33 |
| $\geq 0.5$ | 21 | 143 | 65 | 28 |
| Total | 356 | 356 | 356 | 356 |

considering LST and the prevalence of echinococcosis, except for February and December, all months, spring, summer, autumn, winter and annual average LST were significantly negatively correlated with the prevalence of echinococcosis ($P < 0.05$). In 2018, the minimum value of average NDVI in the overall 370 epidemic counties was 0.1468 in January, gradually increased from January to August to reach the maximum value of 0.4577. It then gradually decreased. A positive rank correlation was found in January and February ($P < 0.05$), both of which were 0.115, while a negative correlation was found in April, May and June ($P < 0.05$).

The lowest elevation among the 370 counties is 178.69 m, in Horqin district, Tongliao City, Inner Mongolia Autonomous Region. The highest elevation is 5155.67 m in Ritu County, Ali Prefecture, Tibet Autonomous Region. Most endemic counties are located between 2000–3000 m (Table 4). Endemic areas in Inner Mongolia, Ningxia and Shaanxi were all of low elevation (less than 3000m) whereas most endemic counties in Tibet were located in high elevation areas. All kind of elevation, *i.e.*, lowest elevation, average elevation and highest elevation were positively correlated ($P < 0.05$) with rank correlation coefficients of 0.568, 0.649 and 0.600, respectively. With respect to correlation analysis between elevation and dog infection rate,

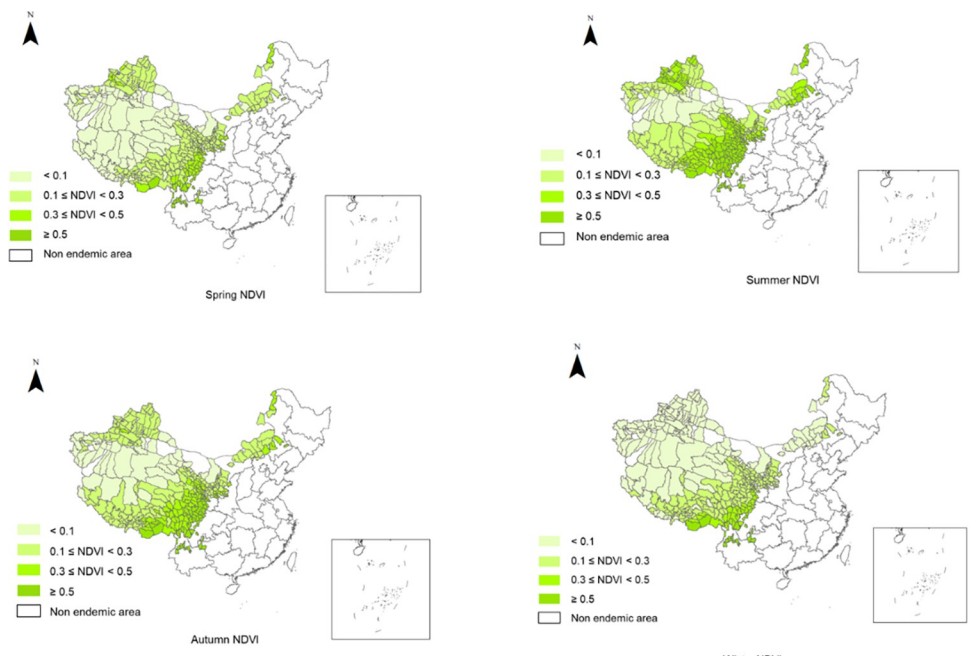

**Fig 3. Seasonal NDVI distribution in endemic counties.** The base layer is from https://www.webmap.cn/mapDataAction.do?method=forw&resType=5&storeId=2&storeName=%E5%9B%BD%E5%AE%B6%E5%9F%BA%E7%A1%80%E5%9C%B0%E7%90%86%E4%BF%A1%E6%81%AF%E4%B8%AD%E5%BF%83&fileId=BA420C422A254198BAA5ABAB9CAAFBC1 with credit to National Catalogue Service For Geographic Information.

**Table 3. Rank correlation between LST, NDVI and prevalence of echinococcosis at county level.**

| LST | r | P | NDVI | r | P |
|---|---|---|---|---|---|
| January LST | -0.176 | 0.001 | January NDVI | 0.115 | 0.031 |
| February LST | 0.031 | 0.562 | February NDVI | 0.115 | 0.029 |
| March LST | -0.536 | 0.000 | March NDVI | -0.038 | 0.473 |
| April LST | -0.575 | 0.000 | April NDVI | -0.186 | 0.000 |
| May LST | -0.613 | 0.000 | May NDVI | -0.202 | 0.000 |
| June LST | -0.569 | 0.000 | June NDVI | -0.112 | 0.034 |
| July LST | -0.576 | 0.000 | July NDVI | -0.008 | 0.879 |
| August LST | -0.583 | 0.000 | August NDVI | -0.009 | 0.866 |
| September LST | -0.575 | 0.000 | September NDVI | -0.079 | 0.138 |
| October LST | -0.599 | 0.000 | October NDVI | -0.055 | 0.297 |
| November LST | -0.353 | 0.000 | November NDVI | -0.055 | 0.298 |
| December LST | -0.085 | 0.111 | December NDVI | 0.062 | 0.241 |
| Spring LST | -0.603 | 0.000 | Spring NDVI | -0.164 | 0.002 |
| Summer LST | -0.585 | 0.000 | Summer NDVI | -0.045 | 0.392 |
| Autumn LST | -0.612 | 0.000 | Autumn NDVI | -0.066 | 0.217 |
| Winter LST | -0.203 | 0.000 | Winter NDVI | 0.049 | 0.360 |
| Annual LST | -0.609 | 0.000 | AnnualNDVI | -0.067 | 0.208 |

lowest, average and highest elevations were all positively correlated ($P < 0.05$) with correlation coefficients of 0.378, 0.38 and 0.306, respectively. There was a positive correlation between the infection rate of Liutiao dogs and the prevalence rate of echinococcosis at the county level by rank correlation analysis with a correlation coefficient of 0.418 ($P < 0.05$).

## Fitting negative binomial model

The mean of human *echinococcus* prevalence (1/100000) in 370 counties was 283.9297779, with a variance of 701.6497667$^2$. Given that the variance was far higher than the mean, the data were over discrete. Thus, we chose negative binomial regression to explore the impact of environmental factors on the prevalence of human echinococcosis. Taking into account the consistent trend of LST across all seasons, which decreases with increasing elevation, we included the average annual LST in the negative binomial regression model. In contrast, due to the inconsistent seasonal trends of NDVI, we incorporated the spring NDVI, summer

**Table 4. Distribution of different elevations in endemic areas.**

| | <1000 | [1000–2000) | [2000–3000) | [3000–4000) | [4000–5000) | >5000 | Total | Prevalence (1/10000) |
|---|---|---|---|---|---|---|---|---|
| Gansu | 0 | 28 | 20 | 9 | 0 | 0 | 57 | 1.55 |
| Inner Mongolia | 8 | 17 | 0 | 0 | 0 | 0 | 25 | 0.69 |
| Ningxia | 0 | 17 | 2 | 0 | 0 | 0 | 19 | 5.42 |
| Qinghai | 0 | 0 | 7 | 18 | 14 | 0 | 39 | 25.63 |
| Shaanxi | 0 | 2 | 0 | 0 | 0 | 0 | 2 | 1.49 |
| Sichuan | 0 | 0 | 4 | 18 | 13 | 0 | 35 | 90.53 |
| Tibet | 0 | 0 | 2 | 1 | 69 | 2 | 74 | 55.40 |
| Xinjiang | 15 | 44 | 17 | 4 | 1 | 0 | 81 | 2.16 |
| Yunnan | 0 | 4 | 17 | 3 | 0 | 0 | 24 | 0.24 |
| Total | 23 | 112 | 69 | 53 | 97 | 2 | 356 | |
| prevalence(1/10000) | 1.4 | 2.3 | 1.5 | 9.4 | 88 | 117.8 | | 10.57 |

**Table 5. Multivariate linear regression fitting of prevalence of human echinococcosis.**

| Variable | Tolerance | VIF | Variable | Tolerance | VIF |
|---|---|---|---|---|---|
| Annual average LST | 0.32863 | 3.04291 | Annual average LST | 0.39657 | 2.5216 |
| Spring NDVI | 0.06596 | 15.16127 | Spring NDVI | 0.2538 | 3.94009 |
| Summer NDVI | 0.0541 | 18.48385 | Summer NDVI | 0.24533 | 4.07616 |
| Autumn NDVI | 0.02041 | 48.99861 | Average elevation | 0.45046 | 2.21996 |
| Winter NDVI | 0.04049 | 24.69935 | Infection rate of dogs | 0.82218 | 1.21628 |
| Average elevation | 0.23804 | 4.20094 | | | |
| Infection rate of dogs | 0.81731 | 1.22353 | | | |

NDVI, autumn NDVI, and winter NDVI separately into the negative binomial regression model. Subsequently, the data of human echinococcosis prevalence (1/100000) was taken as the dependent variable, and the annual mean LST, spring NDVI, summer NDVI, autumn NDVI, winter NDVI, average elevation, and infection rate of dogs determined from an epidemiological survey were taken as independent variables to fit the negative binomial regression model. Firstly, we fitted a multivariate linear regression to check for multicollinearity among the independent variables. If the tolerance level is less than 0.1 or the variance inflation factor (VIF) is more than 10, it signifies severe multicollinearity. As shown in the left column of Table 5, there was severe multicollinearity among the independent variables. As shown in the right column of Table 5, after eliminating certain variables, we were able to solve the problem of multicollinearity.

A negative binomial regression was performed with the data of human echinococcosis prevalence (1/100000) as dependent variable and the remaining environmental factors as independent variables (Table 6).

Among them, annual average LST, spring NDVI, summer NDVI, average elevation and the infection rate of dogs had statistical significance. Prevalence data of human echinococcosis in the 370 counties were over discrete, with a dispersion of 2.822, and the Lagrange multiplier test had no statistical significance, indicating that the data did not follow a Poisson distribution. It was thus reasonable to fit the negative binomial regression model. Logarithmic conversions of meaningful variables are shown in Table 7. The fitting negative binomial regression equation is:

$$\text{Log(Prevalence, } 1/100000) = -6.1754 - 0.1689X1 - 1.8043X2 + 0.0009X3 + 0.0514X4$$

Assuming that other variables remained constant, the mean prevalence of human echinococcosis (1/100000) decreased by 15.5407% when annual average LST increased by 1°C. Similarly, the mean prevalence of human echinococcosis (1/100000) decreased by 83.541% when NDVI increased by one unit in summer. Conversely, the mean prevalence of human echinococcosis (1/100000) increased by 0.090041% with the increase of 1m in average elevation.

**Table 6. Negative binomial regression fitting of human echinococcosis prevalence.**

| Assignment | Independent variable | β | RR | RR 95% CI | | χ2 | P |
|---|---|---|---|---|---|---|---|
| | Intercept | -6.0878 | 0.0023 | 0.0005 | 0.0108 | 58.31 | < .0001 |
| X1 | Annual average LST | -0.1741 | 0.8402 | 0.7732 | 0.9131 | 16.83 | < .0001 |
| X2 | Spring NDVI | 0.3714 | 1.4498 | 0.1675 | 12.5502 | 0.11 | 0.7359 |
| X3 | Summer NDVI | -2.0256 | 0.1319 | 0.0263 | 0.6611 | 6.07 | 0.0138 |
| X4 | Average elevation | 0.0009 | 1.0009 | 1.0007 | 1.0011 | 83.95 | < .0001 |
| X5 | Infection rate of dogs | 0.0495 | 1.0508 | 1.0157 | 1.0871 | 8.18 | 0.0042 |

**Table 7. Negative binomial regression analysis of environmental factors and prevalence of human echinococcosis.**

| Assignment | Independent variable | β | RR | RR 95%CI | | χ2 | P |
|---|---|---|---|---|---|---|---|
| | Intercept | -6.1752 | 0.0021 | 0.0005 | 0.0091 | 67.45 | < .0001 |
| X1 | Annual average LST | -0.1689 | 0.8446 | 0.7816 | 0.9126 | 18.26 | < .0001 |
| X2 | Summer NDVI | -1.8044 | 0.1646 | 0.0619 | 0.4374 | 13.09 | 0.0003 |
| X3 | Average elevation | 0.0009 | 1.0009 | 1.0007 | 1.0011 | 85.05 | < .0001 |
| X4 | Infection rate of dogs | 0.0514 | 1.0528 | 1.0192 | 1.0874 | 9.69 | 0.0018 |

Furthermore, for every 1% increase in the infection rate in dogs, the mean prevalence of human echinococcosis (1/100000) increased by 5.274391%.

By repeating the above steps, we conducted separate negative binomial regression fitting on the prevalence rates of human AE and CE in the 370 counties, respectively. Due to various constraints such as funding, we used the Enzyme-Linked Immunosorbent Assay (ELISA) to detect canine *Echinococcus* infection. However, ELISA can only confirm whether the dogs are infected with *Echinococcus* spp., without distinguishing between *E. granulosus* or *E. multilocularis* infections. Therefore, we did not include the infection rate of dogs in the model when performing negative binomial regression fitting on AE and CE, respectively. For the prevalence of human cystic echinococcosis, we additionally included the prevalence rate of domestic animals into the model. Negative binomial regression was fitted with the data of human CE prevalence (1/100000) as the dependent variable, and the annual average LST, spring NDVI, summer NDVI, autumn NDVI, winter NDVI, average elevation, and the prevalence rate of domestic animals as independent variables. The final results are shown in Table 8. The fitting negative binomial regression equation is:

$$\text{Log(Prevalence, } 1/100000) = -6.7814 - 0.1439X1 + 0.0006X2 + 3.5226X3$$

The annual average LST was inversely correlated with the prevalence of human CE. When other variables were held constant, for each increase of 1°C in the annual mean LST, the mean prevalence of human CE (1/100000) decreased by 13.40256588%. Conversely, both the average elevation and the prevalence rate of domestic animals were positively correlated with the prevalence of human CE.

Negative binomial regression was fitted with the data of human CE prevalence (91/100000) as the dependent variable, and the annual average LST, spring NDVI, summer NDVI, autumn NDVI, winter NDVI, and average elevation as independent variables. Results are shown in Table 9. The fitting negative binomial regression equation is:

$$\text{Log(Prevalence, } 1/100000) = -15.8432 + 13.8871X1 - 22.2975X2 + 0.0023X3$$

The average elevation was positively correlated with the prevalence of human AE. When all other variables were kept constant, for each increase of one meter in the average elevation, the mean prevalence of human CE (1/100000) increased by 0.230264703 percent. The NDVI was

**Table 8. Negative binomial regression analysis of environmental factors and prevalence of human cystic echinococcosis.**

| Assignment | Independent variable | β | RR | RR 95% CI | | χ² | P |
|---|---|---|---|---|---|---|---|
| | Intercept | -6.7814 | 0.0011 | 0.00025 | 0.005 | 76.32 | < .0001 |
| X1 | Annual average LST | -0.1439 | 0.866 | 0.79538 | 0.943 | 11.01 | 0.0009 |
| X2 | Average elevation | 0.0006 | 1.0006 | 1.00033 | 1.001 | 19.17 | < .0001 |
| X3 | The prevalence rate of domestic animals | 3.5226 | 33.873 | 1.69313 | 677.666 | 5.31 | 0.0212 |

**Table 9. Negative binomial regression analysis of environmental factors and prevalence of human alveolar echinococcosis.**

| Assignment | Independent variable | B | RR | RR 95% CI | | $\chi^2$ | P |
|---|---|---|---|---|---|---|---|
|  | Intercept | 15.8432 | 0.0000001363 | 0.00000001914 | 0.0000009052 | 259.37 | < .0001 |
| X1 | Autumn NDVI | 13.8871 | 1074203.8308 | 1651.2955 | 698793090.08 | 17.65 | < .0001 |
| X2 | Winter NDVI | 22.2975 | 0.00000000021 | 0 | 0.0000439 | 12.70 | 0.0004 |
| X3 | Average elevation | 0.0023 | 1.0023 | 1.0018 | 1.0027 | 96.08 | < .0001 |

positively correlated with human AE prevalence in autumn, whereas it was inversely correlated with human AE prevalence in winter.

## Discussion

*Echinococcu*s requires two mammalian hosts to complete its life cycle [5]. The definitive host, dogs, excretes feces containing *Echinococcus* eggs which pollute local water sources, food and pasture. The intermediate hosts of *Echinococcus*, livestock, are infected during grazing. The viscera of infected livestock are fed to dogs, and the parasite develops into the adult stage in dogs to initiate a spreading cycle. Host transmission of *E. multilocularis* occurs between stray dogs or foxes as definitive host and small rodents such as voles or pikas as intermediate hosts [17]. The spatial overlap and predation relationship between the definitive host and the intermediate host are related to landscape factors, which can directly affect the spread of *Echinococcus*. Elevation, LST and NDVI are the main landscape factors affecting the prevalence of echinococcosis [12,18–19].

Temperature and humidity are the main determinants of the survival rate of parasite eggs in the environment [20–21]. *Echinococcus* eggs are sensitive to high temperature but resistant to cold. Areas with low LST are more likely to be infected with echinococcosis [22]. LST reflects the change of temperature in the environment [12]. Our results indicated a significant negative correlation between the prevalence of human echinococcosis and the mean LST for each season, as well as the annual average LST. In the negative binomial regression model, annual average LST was significantly negatively correlated with the prevalence of human echinococcosis. Similarly, the prevalence of human CE also showed a significant negative correlation with the annual average LST. Some studies have found a negative correlation between Spring LST and CE prevalence [23]. It shows that LST is the main environmental factor affecting the prevalence of echinococcosis in the population. The lower the LST, the higher the risk for the population. Temperature also affects geographical distribution and changes the composition of small mammal communities [24–25]. Climate has been identified as a factor leading to changes in the distribution and number of red foxes, which are the definitive hosts of *E. multilocularis* [26]. In Western China, the elevation is high with a typical plateau and mountain climate where the LST is kept low all year round. Climatic conditions are very conducive to the survival of *echinococcus* eggs [6]. Early winter and early spring are high incidence seasons for dogs infected with *Echinococcus*. During the traditional Chinese Spring Festival, the number of slaughtered livestock increases making domestic dogs more likely to be in contact with viscera [27]. Extreme cold weather often occurs in this season, which may cause the intermediate host animals to freeze to death in the wild, directly increasing the field transmission cycle.

In our work, the average elevation was positively correlated with the prevalence of human echinococcosis, whether AE or CE. This confirms previous studies [6,8,18]. The higher the elevation, especially above 3000 meters, the smaller the proportion of agricultural production, while the proportion of animal husbandry increases, with grasslands gradually replacing

farmland. The area with large grassland proportion not only increases the number of intermediate hosts and livestock by provides a suitable environment. At the same time, the number of definitive hosts such as dogs also increases sharply. Herdsmen are raising dogs to protect livestock during grazing. Together, all these factors contribute to a significant increase in the total number of hosts and strengthen the spreading cycle of *Echinococcus*.

Previous studies on echinococcosis have confirmed that the change of land cover is related to the increase of population density of the critical intermediate host of *Echinococcus* [28]. NDVI is an important factor affecting the animal host distribution of *Echinococcus* [29]. The winter prevalence of echinococcosis in Western China is positively correlated with NDVI [23]. However, in winter, most of the high elevation areas with a concentrated number of cases are covered by snow and ice for extended periods, resulting in low NDVI in these areas. Therefore, there is no significant relationship between NDVI in winter or December and the prevalence of human echinococcosis in this study. In summer, pastures lead to an increase of NDVI values. In autumn, as the temperature decreases, pastures and farmlands become barren which is not conducive to the growth of small rodents and the grazing behavior of livestock. Grassland vegetation coverage will directly affect the distribution of intermediate hosts, and livestock play an important ecological role in the spread of CE [28]. Grassland is one of the living conditions for the growth and reproduction of horses, cattle, sheep and other intermediate hosts [29–30]. In pastoral areas, when yaks, sheep, horses and other livestock jointly graze, the grassland coverage and vegetation height will be reduced, which makes the population density of plateau pika larger than that of natural grassland [31]. There is a significant positive correlation between the prevalence of AE and the forest, grassland and shrub vegetation near villages and a negative correlation with the cultivated land area [32–33]. A study in France showed that the population of voles in these areas erupted periodically and the population density increased sharply [30]. It was also reported in China that the prevalence of AE in the human population increased in areas with a high proportion of meadows while very few cases have been found in areas with poor vegetation (marshland) [6]. Lowland pastures are described as heavily grazed pastures scattered with forest or shrub cover, which is related to a high human incidence rate [10].

In 1999, the project of returning farmland to forests was implemented to restore the previous ecological environment through three types of land transformation, namely, farmland to grassland, farmland to forest and wasteland to forest. These changes in land cover are likely to promote the spread of *E. multilocularis*, which will increase the density and distribution of small mammals [34]. With the process of deforestation, the increase of grasslands or shrubs is conducive to the creation of near domestic habitats for small mammals and the development of near domestic cycles involving dogs. The distribution of small mammals in Gansu Province is also due to the short-term increase of grasslands and shrubs after deforestation [27,35]. In eastern France, voles and vole population outbreaks have been reported in areas where cultivated land has been converted to permanent grassland [28]. Reforested lowland pastures are by definition covered with forests or shrubs leading to a higher prevalence of human AE [10]. A study in Ningxia Hui Autonomous Region showed that the abundance of degraded lowland pastures is related to the higher prevalence of AE [10]. Landscape features may directly or indirectly determine the feeding behavior, growth rate, reproductive efficiency and immune mechanism of livestock [36]. Grazing or trampling will affect the quality of grassland and the length of forage grass, which will provide a better habitat for small mammals [37].

The geographical distribution of echinococcosis in China is uneven. The Qinghai Tibet Plateau is a hot spot of echinococcosis epidemic in China, with an average elevation of more than 4000 m [4]. The Qinghai Tibet Plateau is generally a dry and cold tundra, but there are differences in different regions. The annual precipitation in the western region is less than 100 mm,

making it dry and sparsely populated, while the annual precipitation in the eastern region is 500–700 mm and thus a greater livestock with a higher population density can be maintained [30]. The cultivated area in the Qinghai Tibet Plateau accounts for less than 1% of the surface and is restricted to regions lower than 3500 m. The main vegetation is grass and sedge (excluding Carex, Ceratoides, Ferns and Kobresia) and pastures account for most of the area, including alpine meadows and alpine grasslands, with a total surface of 8.7 million hectares [38]. These natural conditions lead to a higher prevalence of echinococcosis in the Qinghai Tibet Plateau. This region is mostly devoted animal husbandry and the socio-economic situation is relatively lagging. Basic healthcare is far from perfect and the awareness about disease prevention is relatively poor. Moreover, increased host range and enhanced parasitic transmission between definitive and intermediate hosts, caused by environmental changes, might put humans at risk of increased echinococcosis transmission [39]. Collectively, these factors contribute to the spatial distribution characteristics of echinococcosis. Additional knowledge about how environmental conditions affect *E. granulosus* transmission would be helpful in planning CE control and management programs [40]. Since dogs are the definitive hosts of *E. granulosus* and *E. multilocularis* [41] and humans are accidental intermediate hosts ingesting contaminated food or water [42], domestic dogs and stray dogs in villages make the primary source of infection [43].

Firstly, this study explored the environmental risk factors of human echinococcosis at the county level, which is a relatively broad scale. In the future, if possible, we can delve deeper by conducting research at the township or even village level. Secondly, the incubation period, as a confounding factor, might have some impact on the results of this study. However, considering that environmental factors don't vary greatly in the same region across different years, our results are still reasonable. Thirdly, fecal ELISA detection technology lacks specificity, using ELISA to determine the infection rate in dogs has its limitations. In the future, further PCR testing should be conducted on dogs that tested positive with ELISA to determine whether they are infected by *E. granulosus* or *E. multilocularis*.

## Conclusion

This article addressed the impact of environmental factors on population prevalence in all endemic counties. The prevalence of human hydatid disease is affected by environmental factors, such as annual average LST, spring NDVI, summer NDVI, average elevation. It is necessary to pay special attention to these areas, strengthen environmental risk monitoring, carry out targeted prevention and control, and rationally allocate health resources. We cannot reduce the prevalence of echinococcosis by intervening in environmental factors. But our research can suggest in which environments the prevalence of echinococcosis will be more severe, thus focusing on prevention and control.

## Acknowledgments

Since the national echinococcosis control project was launched in 2006, it has been supported by peers in all epidemic areas. We sincerely thank all the participants who participated in the prevention and control activities in the 370 echinococcosis endemic counties in 9 epidemic provinces (plus Xinjiang production and Construction Corps).

## Author Contributions

**Conceptualization:** Liying Wang, Jun Yan, Roger Frutos.

**Data curation:** Liying Wang, Min Qin, Jun Yan.

**Formal analysis:** Liying Wang, Zhiyi Wang, Min Qin, Jiaxi Lei.

**Funding acquisition:** Liying Wang.

**Investigation:** Liying Wang, Jun Yan.

**Methodology:** Liying Wang, Jun Yan, Laurent Gavotte, Roger Frutos.

**Project administration:** Liying Wang.

**Resources:** Liying Wang, Jun Yan.

**Software:** Liying Wang, Zhiyi Wang, Min Qin, Jiaxi Lei.

**Supervision:** Jun Yan, Laurent Gavotte, Roger Frutos.

**Validation:** Liying Wang, Laurent Gavotte, Roger Frutos.

**Visualization:** Min Qin, Jiaxi Lei, Xixi Cheng.

**Writing – original draft:** Liying Wang, Zhiyi Wang, Min Qin, Jiaxi Lei.

**Writing – review & editing:** Liying Wang, Jun Yan, Laurent Gavotte, Roger Frutos.

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
