## [Decision Letter · Decision Letter 0]

13 Dec 2023

Dear Professor WANG,

Thank you very much for submitting your manuscript "Study on the main environmental risk factors of human echinococcosis at 370 counties, China" for consideration at PLOS Neglected Tropical Diseases. As with all papers reviewed by the journal, your manuscript was reviewed by members of the editorial board and by several independent reviewers. In light of the reviews (below this email), we would like to invite the resubmission of a significantly-revised version that takes into account the reviewers' comments. 

We cannot make any decision about publication until we have seen the revised manuscript and your response to the reviewers' comments. Your revised manuscript is also likely to be sent to reviewers for further evaluation.

Sincerely,

María Victoria Periago

Academic Editor

Aaron Jex

Section Editor

Reviewer's Responses to Questions

**Key Review Criteria Required for Acceptance?**

**Methods**

-Are the objectives of the study clearly articulated with a clear testable hypothesis stated?

-Is the study design appropriate to address the stated objectives?

-Is the population clearly described and appropriate for the hypothesis being tested?

-Is the sample size sufficient to ensure adequate power to address the hypothesis being tested?

-Were correct statistical analysis used to support conclusions?

-Are there concerns about ethical or regulatory requirements being met?

Reviewer #1: -Are the objectives of the study clearly articulated with a clear testable hypothesis stated?

Yes

-Is the study design appropriate to address the stated objectives?

It is necessary to detail the methodology used for diagnosis in dogs. Environmental samples or samples collected directly from the host were analyzed?

The diagnostic methodology to estimate the prevalence in humans is not described. It is very important to discriminate the age range, since cases in people under 15 years of age refer to active cycles of CE transmission.

-Is the population clearly described and appropriate for the hypothesis being tested?

yes

-Is the sample size sufficient to ensure adequate power to address the hypothesis being tested?

yes

-Were correct statistical analysis used to support conclusions?

yes

-Are there concerns about ethical or regulatory requirements being met?

yes

Reviewer #2: 1. The authors have stated in the Introduction that the prevalence of echinococcosis and environmental factors have strong spatial autocorrelation. But when determining the relationship of environmental factors and the risk of echinococcosis in negative binomial regression model, the spatial autocorrelation is not well considered and controlled in model fitting.

2. Reasons of LST, NDVI, and elevations groupings are not described. 

3. Social factors are also significantly involved in the epidemic process of human echinococcosis, and why it is not controlled here?

**Results**

-Does the analysis presented match the analysis plan?

-Are the results clearly and completely presented?

-Are the figures (Tables, Images) of sufficient quality for clarity?

Reviewer #1: -Does the analysis presented match the analysis plan?

The samples analyzed by ELISA were confirmed by WB? If so, these results must be reported. It is known that there are cross reactions with other helminths, this denotes a strong limitation for the interpretation of the results, due to the possibility of false positives.

-Are the results clearly and completely presented?

Yes

-Are the figures (Tables, Images) of sufficient quality for clarity?

Yes

Reviewer #2: Betas and its 95% CIs from regression models are insufficient for public health practitioners to get the epidemiological meaning of the association in magnitude, and I would like to suggest to use its exponential form here.

**Conclusions**

-Are the conclusions supported by the data presented?

-Are the limitations of analysis clearly described?

-Do the authors discuss how these data can be helpful to advance our understanding of the topic under study?

-Is public health relevance addressed?

Reviewer #1: "Extreme cold weather often occurs in this season, which may cause the

intermediate host animals to freeze to death in the wild, directly increasing the field

transmission cycle". 

This can also imply the death of the metacestode

"Studies have found that when NDVI is below

0.5 or above 0.7 in spring, the prevalence of echinococcosis gradually increases, and

when NDVI is between 0.5 and 0.7, the prevalence gradually decreases"

Please explain better the cause of this variation.

The authors should discuss the lack of specificity of the technique used to detect cases in definitive hosts (copro-ELISA). Although this work provides valuable information on the epidemiology landscape, the concept of one health is not presented, addressed or discussed.

Due to there are cross-reactions with other taenids, how can you ensure that positive results only belong to echinococcus spp.?

Reviewer #2: The authors concluded that the main environmental risk factors were altitude, LST, and NDVI associated with echinococcosis, which was not sufficiently supported by the results.

**Editorial and Data Presentation Modifications?**

Reviewer #1: The manuscript needs severe revisions regarding terminology. It is mandatory to unify terms such as: definitive host, intermediate host, echinococcus spp. eggs, E. granulosus s.l., etc. I suggest consulting "International consensus on terminology to be used in the field

of echinococcoses" Vuitton et al., 2020

https://doi.org/10.1051/parasite/2020024

Reviewer #2: The tables and figures need arrangement and modifications to make the results more concise.

**Summary and General Comments**

Reviewer #1: This work describes the relationships between environmental elements and the risk of human echinococcosis. Although this work responds to a one-health approach, this factor is not adequately addressed. The manuscript needs a thorough review of language and terminology. The authors must report limitations when calculating the prevalence in definitive hosts, due to the lack of specificity of the technique used (copro-ELISA). Likewise, it is necessary to provide more information on how the prevalence data in humans were obtained.

Reviewer #2: Present work investigated several environmental factors and its association with the prevalence of human echinococcosis in epidemic areas of mainland China. In reviewing the work, several potential limitations and deficiencies in terms of study design come to attention. It is necessary to address these issues to enhance the rigor and reliability of the study's findings.

Findings were difficult to adjudicate due to the poor grammar and structure problem. The paper needs English language professional editing. And laborious and problematic sentences are not limited to the following sentences: “40% of CE cases and 90% of AE cases worldwide occur in China and the disease burden of CE and AE in China respectively accounts for 40% and 90% of the global disability-adjusted life years (DALYs)” in page 8 and “The lower the LST, the higher the risk of the population” in page 21. 

The current title was not self-explanatory and its study design are recommended to added.

PLOS authors have the option to publish the peer review history of their article (what does this mean?). If published, this will include your full peer review and any attached files.

Reviewer #1: Yes: Héctor Gabriel Avila

Reviewer #2: No
---

## [Decision Letter · Decision Letter 1]

31 Mar 2024

Dear Professor WANG,

We are pleased to inform you that your manuscript 'A regressive analysis of the main environmental risk factors of human echinococcosis in 370 counties in China' has been provisionally accepted for publication in PLOS Neglected Tropical Diseases.

Best regards,

María Victoria Periago

Academic Editor

Richard Bradbury

Section Editor

Please make sure to include the reference suggested by the reviewer and also make a thorough review of the manuscript for typing errors.

Reviewer's Responses to Questions

**Key Review Criteria Required for Acceptance?**

**Methods**

-Are the objectives of the study clearly articulated with a clear testable hypothesis stated?

-Is the study design appropriate to address the stated objectives?

-Is the population clearly described and appropriate for the hypothesis being tested?

-Is the sample size sufficient to ensure adequate power to address the hypothesis being tested?

-Were correct statistical analysis used to support conclusions?

-Are there concerns about ethical or regulatory requirements being met?

Reviewer #1: Yes

**Results**

-Does the analysis presented match the analysis plan?

-Are the results clearly and completely presented?

-Are the figures (Tables, Images) of sufficient quality for clarity?

Reviewer #1: Yes

**Conclusions**

-Are the conclusions supported by the data presented?

-Are the limitations of analysis clearly described?

-Do the authors discuss how these data can be helpful to advance our understanding of the topic under study?

-Is public health relevance addressed?

Reviewer #1: Yes

**Editorial and Data Presentation Modifications?**

Reviewer #1: "Extreme cold weather often occurs in this season, which may cause the intermediate host animals

to freeze to death in the wild, directly increasing the field transmission cycle", add cite.

**Summary and General Comments**

Reviewer #1: -

PLOS authors have the option to publish the peer review history of their article (what does this mean?). If published, this will include your full peer review and any attached files.

Reviewer #1: **Yes: **Héctor Gabriel Avila

---

## [Editor Report · Acceptance letter]

3 May 2024

Dear Professor WANG,

We are delighted to inform you that your manuscript, "A regressive analysis of the main environmental risk factors of human echinococcosis in 370 counties in China," has been formally accepted for publication in PLOS Neglected Tropical Diseases.

Best regards,

Shaden Kamhawi

co-Editor-in-Chief

Paul Brindley

co-Editor-in-Chief
